# Patlak Slope versus Standardized Uptake Value Image Quality in an Oncologic PET/CT Population: A Prospective Cross-Sectional Study

**DOI:** 10.3390/diagnostics14090883

**Published:** 2024-04-24

**Authors:** Semra Ince, Richard Laforest, Malak Itani, Vikas Prasad, Saeed Ashrafinia, Anne M. Smith, Richard L. Wahl, Tyler J. Fraum

**Affiliations:** 1Department of Radiology, Washington University School of Medicine, 510 S. Kingshighway Blvd, Campus Box 8131, St. Louis, MO 63110, USA; drsemra@gmail.com (S.I.); rlaforest@wustl.edu (R.L.); mitani@wustl.edu (M.I.); pvikas@wustl.edu (V.P.); rwahl@wustl.edu (R.L.W.); 2Siemens Medical Solutions USA, Inc., 810 Innovation Drive, Knoxville, TN 37932, USA; saeed.ashrafinia@siemens-healthineers.com (S.A.); anne.m.smith@siemens-healthineers.com (A.M.S.); 3Department of Radiation Oncology, Washington University School of Medicine, 660 S. Euclid Ave, MSC 8224-35-LL, St. Louis, MO 63110, USA

**Keywords:** Patlak slope, PET, SUV, metabolic rate, oncology

## Abstract

Patlak slope (PS) images have the potential to improve lesion conspicuity compared with standardized uptake value (SUV) images but may be more artifact-prone. This study compared PS versus SUV image quality and hepatic tumor-to-background ratios (TBRs) at matched time points. Early and late SUV and PS images were reconstructed from dynamic positron emission tomography (PET) data. Two independent, blinded readers scored image quality metrics (a four-point Likert scale) and counted tracer-avid lesions. Hepatic lesions and parenchyma were segmented and quantitatively analyzed. Differences were assessed via the Wilcoxon signed-rank test (alpha, 0.05). Forty-three subjects were included. For overall quality and lesion detection, early PS images were significantly inferior to other reconstructions. For overall quality, late PS images (reader 1 [R1]: 3.95, reader 2 [R2]: 3.95) were similar (*p* > 0.05) to early SUV images (R1: 3.88, R2: 3.84) but slightly superior (*p* ≤ 0.002) to late SUV images (R1: 2.97, R2: 3.44). For lesion detection, late PS images were slightly inferior to late SUV images (R1 only) but slightly superior to early SUV images (both readers). PS-based TBRs were significantly higher than SUV-based TBRs at the early time point, with opposite findings at the late time point. In conclusion, late PS images are similar to early/late SUV images in image quality and lesion detection; the superiority of SUV versus PS hepatic TBRs is time-dependent.

## 1. Introduction

The standardized uptake value (SUV) is the most widely utilized quantitative metric in positron emission tomography (PET) [1]. In oncologic imaging, the vast majority of image reconstructions are based on the SUV, which reflects the number of decay events in a given volume of tissue, regardless of whether that tracer is bound or unbound. As such, physiologic tracer uptake by normal organs on SUV images can obscure tracer-avid lesions, resulting in inaccurate tumor burden assessments [2]. The Patlak model attempts to address this shortcoming [3]. This model assumes that the circulating tracer is trapped irreversibly, allowing the tracer’s net uptake rate to be estimated via the Patlak slope (PS) [4]. Several clinically utilized oncologic PET tracers generally exhibit this behavior, thereby allowing Patlak modeling of dynamic whole-body (WB) PET data [5,6]. Importantly, PS images, by removing the signal derived from the unbound tracer, have the potential to improve lesion conspicuity in organs with relatively high background parenchymal activity (e.g., liver) [7].

Despite many publications on the Patlak method, only two have evaluated the clinical utility of WB PS images [8,9]. These studies reported fewer false positive and false negative findings and better tumor-to-background ratios (TBRs) on PS images relative to SUV images but did not utilize identical post-injection intervals for SUV and PS reconstructions. As such, disparate effective uptake times may have contributed to differences in SUV versus PS image quality. Thus, the aim of our study was to compare the visual quality and hepatic TBRs of SUV images to PS images at equivalent post-injection time points.

## 2. Materials and Methods

### 2.1. Study Design

This prospective study, which occurred at a single tertiary care center, was approved by our local Institutional Review Board and complied with the standards of the Health Insurance Portability and Accountability Act. We enrolled subjects already scheduled to undergo standard-of-care (SOC) oncologic PET/computed tomography (CT) examinations with one of the following tracers, all of which satisfy the assumptions of the Patlak model [5,6,10,11]: 2-deoxy-2-[^18^F]fluoro-D-glucose ([^18^F]FDG); [^68^Ga]Ga-DOTATATE or [^64^Cu]Cu-DOTATATE (hereafter collectively called DOTATATE, given our analytic pooling of ^64^Cu and ^68^Ga cases); or [^18^F]piflufolastat ([^18^F]DCFPyL). The inclusion criteria were as follows: age of 18 years or older; ability to provide written informed consent; and ability (self-reported) to undergo approximately 90 min of supine imaging with minimal motion. Informed consent was obtained from all subjects involved in the study. Study imaging occurred immediately prior to and following the SOC PET/CT acquisition in a single session.

### 2.2. Imaging Protocol

The imaging protocol is captured in Figure 1. All study imaging occurred on a single Biograph Vision 600 PET/CT scanner (Siemens Healthineers; Knoxville, TN, USA) utilizing Food and Drug Administration-approved, commercially available software for on-line reconstruction of multiparametric PET images (FlowMotion Multiparametric PET Suite; Siemens Healthineers; Knoxville, TN, USA). Subjects undergoing [^18^F]FDG imaging were required to fast for at least 4 h prior to [^18^F]FDG injection; a blood glucose of 200 mg/dL or less was required at the time of tracer administration. [^18^F]FDG dosing followed a weight-based schema: <54 kg–370 MBq; 55–113 kg—555 MBq; >113 kg—740 MBq. [^68^Ga]Ga-DOTATATE dosing was also weight-based: 2.0 MBq/kg. In contrast, [^64^Cu]Cu-DOTATATE (333 MBq) and [^18^F]DCFPyL (148 MBq) doses were identical across all weights. The tracer was injected intravenously, with the patient positioned supine within the scanner bore. A 6-min dynamic PET acquisition, centered about the heart, was performed, ensuring that the initial intravascular bolus arrival was captured. The subsequent variable-duration ‘whole-body’ PET passes utilized continuous bed motion and list mode acquisition. Five 2-min passes, followed by five 5-min passes, were performed before SOC imaging. Three additional 5-min passes were performed after SOC imaging. The craniocaudal range of these ‘whole-body’ passes was determined by the clinical indication, most commonly extending from the skull base through the proximal thighs. Note that subjects left the scanner table to empty their urinary bladder immediately before the SOC acquisition. Both low-dose CT scans utilized the following parameters: CARE Dose4D—111 mAs (reference); CARE kV—120 kV (reference); ADMIRE strength of 2.

### 2.3. PET Image Reconstruction

Cylindrical volumes of interest (VOIs) were automatically placed by the scanner software in the descending thoracic aorta on all PET acquisitions. Blood activity concentrations were extracted at each time point to derive a time-activity curve (i.e., arterial input function), as required for Patlak modeling [12]. Prior to Patlak reconstruction, the ‘whole-body’ passes were dynamically reviewed by one of the study investigators to ensure that the images were not substantially degraded by bulk body motion. SUV and PS image reconstructions utilized the following parameters per manufacturer recommendations: SUV—time-of-flight, point-spread-function, 4 iterations, 5 subsets, 440 × 440 matrix, all-pass filter; PS—time-of-flight, point-spread-function, 8 iterations, 5 subsets, 220 × 220 matrix, 2 mm Gaussian filter. Note that PS images are intrinsically noisier than SUV images and that the differences in these parameters (e.g., smaller matrix size) were intended to achieve a similar level of image noise across reconstructions. Time-matched PS and SUV images were reconstructed from three 5-min ‘whole-body’ passes performed approximately 35–50 min (early) or 75–90 min (late) following tracer injection. We utilized the three latest passes before the SOC imaging for reconstruction of the early images to ensure adequate time for steady-state conditions to be achieved.

The SUV calculation utilized actual body weight; SUV had units of g/mL. For [^18^F]FDG studies, PS had units of mg/min/100 mL, as the scanner-derived PS values were multiplied by the patient’s blood glucose level at the time of tracer injection; this approach accounts for the effects of large differences in blood glucose levels on the rate of irreversible [^18^F]FDG trapping. This version of the PS is equivalent to the metabolic rate of [^18^F]FDG (MR_FDG_). For DOTATATE or [^18^F]DCFPyL studies, PS had units of ml/min/100 mL. This version of the PS is equivalent to the influx constant (Ki). The MR_FDG_ and the Ki are collectively called the PS in this study. Additionally, as DOTATATE and [^18^F]DCFPyL remain in the blood plasma (i.e., do not equilibrate with the red blood cell cytoplasm like [^18^F]FDG), the PS values for cases performed with these tracers were retrospectively corrected for the patient’s hematocrit as follows [13,14]:$$corrected\;PS=\frac{measured\;PS}{1-hematocrit}$$

### 2.4. Quantitative Analysis

Tracer-avid hepatic lesions felt to represent sites of malignancy on the SOC PET/CT interpretation were identified. In the case of numerous lesions, the largest and/or most tracer-avid (up to a maximum of 5 per patient) were selected. Utilizing co-registered CT images for guidance, each lesion was manually segmented in MIM version 7.1.5 (MIM Software; Cleveland, OH, USA) on four PET image sets (PS-early, SUV-early, PS-late, SUV-late), thereby generating 4 separate VOIs for each lesion. Maximum (max) and peak values were extracted for each lesion [1]. Additionally, for each PET reconstruction, a spherical VOI of 3 cm diameter was placed in the right hemiliver (avoiding areas of pathology) to extract early and late mean hepatic SUVs and PS values. Early and late TBRs, defined as the ratio of a lesion’s maximum or peak value to the background liver mean value, were calculated.

### 2.5. Qualitative Analysis

Two independent readers blinded to reconstruction type assessed each PET image set with co-registered CT images for a given participant in a single session. The assessment order was randomized on a per-subject basis to mitigate the systematic effects of recall bias. Overall image quality, image noise, artifact freeness, and lesion conspicuity were scored via a 4-point Likert scale (1 = worst; 4 = best). Furthermore, readers recorded the number of presumably malignant tracer-avid lesions for each reconstruction, using the reconstruction with the fewest such lesions as the reference (i.e., relative lesion number). For a given subject, the reconstruction with the fewest lesions was assigned a score of 0. The other three reconstructions were assigned a number indicating how many more lesions were apparent on that reconstruction. For example, if the early PS images showed 8 lesions, the early SUV images showed 9 lesions, and the late PS and late SUV images each showed 11 lesions, the relative lesion number was scored as follows: early PS—0; early SUV—1; late PS—3; late SUV—3.

### 2.6. Statistical Analysis

Prism 9 (GraphPad Software; San Diego, CA, USA) and Excel 2016 (Microsoft, Inc.; Redmond, WA, USA) were utilized for statistical analysis. Demographic, oncologic, and PET/CT characteristics were summarized descriptively. Pairwise comparisons of quantitative and qualitative variables, many of which were deemed to be non-normal via the Shapiro-Wilk test, were performed via the two-tailed Wilcoxon signed-rank test. Separate analyses were performed for all cases, for the [^18^F]FDG subgroup, and for the DOTATATE subgroup. There were insufficient [^18^F]DCFPyL cases for subgroup analysis. To assess qualitative inter-reader agreement for the qualitative analysis, we utilized percent agreement rather than kappa due to multiple instances in which both readers preferred the same reconstruction for all or nearly all cases. As a result, kappa could not be estimated. For a given pairwise comparison, percent agreement was defined as the number of cases in which (A) both readers preferred the same reconstruction (regardless of magnitude) or (B) both readers had a lack of preference, divided by the total number of cases (n = 43). A 95% confidence interval (CI) for the percent agreement was calculated via binomial exact proportions due to the relatively small sample size. *p* < 0.05 defined statistical significance.

## 3. Results

### 3.1. Study Cohort

Seventy-eight patients were enrolled in the study. Forty-three subjects (33 [^18^F]FDG, 8 DOTATATE, and 2 [^18^F]DCFPyL) were deemed to have tracer-avid, presumably malignant lesions on the clinical reports for the SOC portions of their PET/CT examinations and were included in the analysis (Figure 2). This study cohort was 60.5% male (26/43) with a mean age of 63.3 years. Additional patient and scan characteristics are summarized in Table 1.

### 3.2. Image Quality of SUV vs. PS Reconstructions

Appendix A show the results of the qualitative analyses across all tracers and for both subgroups. These data are also summarized visually for all tracers (Figure 3), as well as for the [^18^F]FDG (Figure 4) and DOTATATE (Figure 5) subgroups. An example case is shown in Figure 6. The values in this subsection are means.

For overall image quality (R1: 3.95 vs. 1.19; R2: 3.95 vs. 2.14), image noise, and artifact freeness, both readers rated the PS-early images as inferior (all *p* values < 0.001) to the other reconstructions across all tracers, with high agreement (range: 81.4–100%). PS-early images also had significantly lower lesion conspicuity than the other reconstructions for both readers, though with lower agreement (range: 44.2–69.8%). PS-early images had significantly fewer tracer-avid lesions relative to the other reconstructions for both readers (with the exception of reader 1 when comparing with SUV-early images). The [^18^F]FDG subgroup analysis produced similar results. The DOTATATE subgroup analysis had too few lesions to assess the relative lesion number statistically.

The relationships among the PS-late, SUV-early, and SUV-late reconstructions were more heterogeneous. For example, across all tracers, both readers preferred PS-late images to SUV-late images (*p* ≤ 0.002) for overall image quality, image noise, and artifact freeness, though some of these relationships did not persist for reader 2 in the subgroup analysis. However, across all tracers, there were no significant differences between PS-late images and SUV-late images in terms of lesion conspicuity and relative lesion number, with the exception of a slightly higher relative lesion number for reader 1 on the SUV-late images (2.09 vs. 1.35; *p* = 0.04). For overall image quality and image noise (across all tracers), both readers preferred SUV-early images to SUV-late images (all *p* ≤ 0.03), but with different results for each reader for the other qualitative features. Finally, across all tracers and for both subgroups, SUV-early and PS-late images scored similarly in overall image quality and image noise; however, both readers reported significantly higher lesion conspicuity and relative lesion number for PS-late images across all tracers and for the [^18^F]FDG subgroup (*p* ≤ 0.02).

### 3.3. Hepatic TBRs on SUV vs. PS Images

Among the 43 subjects included in the qualitative analysis, 15 subjects (7 [^18^F]FDG, 8 DOTATATE) had a total of 36 tracer-avid liver lesions (18 [^18^F]FDG, 18 DOTATATE). Table 2 shows the results of the TBR analysis. Values in this subsection are medians. Across all tracers, hepatic TBRs were slightly but significantly higher at the early versus late time point when based on PS-max (3.87 vs. 3.57; *p* < 0.001) and PS-peak (2.90 vs. 2.80; *p* = 0.03), though with opposite trends for the [^18^F]FDG and DOTATATE subgroups. In contrast, across all tracers, hepatic TBRs were significantly lower at the early versus late time point when based on SUV-max (3.09 vs. 5.29; *p* < 0.001) and SUV-peak (2.28 vs. 3.10; *p* < 0.001), with mostly similar findings in the subgroup analyses.

Across all tracers, hepatic TBRs were significantly higher at the early time point for PS-max versus SUV-max (3.87 vs. 3.09; *p* = 0.006) and for PS-peak versus SUV-peak (2.90 vs. 2.28; *p* = 0.003). Similar findings were observed for the [^18^F]FDG subgroup; however, for the DOTATATE subgroup, there were no significant differences in early TBRs between SUV-based metrics and PS-based metrics. In contrast, across all tracers, hepatic TBRs were significantly lower at the late time point for PS-max versus SUV-max (3.57 vs. 5.29; *p* < 0.001) and for PS-peak versus SUV-peak (2.80 vs. 3.10; *p* < 0.001). Equivalent, statistically significant late hepatic TBR findings were also observed for the [^18^F]FDG and DOTATATE subgroups. An example case is shown in Figure 7.

## 4. Discussion

In this study, we examined PS versus SUV overall image quality and found that late PS images (R1: 3.95, R2: 3.95) were similar (*p* > 0.05) to early SUV images (R1: 3.88, R2: 3.84) but slightly superior (*p* ≤ 0.002) to late SUV images (R1: 2.97, R2: 3.44), with more pronounced superiority (*p* < 0.001) relative to early PS images (R1: 1.19, R2: 2.14). In terms of relative lesion number, late PS images outperformed early SUV images for both readers but were slightly inferior to late SUV images for one reader only; again, early PS images were generally inferior to other reconstructions. Finally, among hepatic lesions, early TBRs were higher for PS images, whereas late TBRs were higher for SUV images.

Our finding that late PS images are similar (or sometimes superior) to SUV images in terms of multiple qualitative metrics agrees with previously published data. For example, a study of 18 patients undergoing oncologic [^18^F]FDG-PET/CT found that PS images were of similar or slightly inferior image quality relative to SUV images [9]. However, PS images were reconstructed from earlier post-injection time points than SUV images, possibly contributing to lower PS image quality. A similar study of 109 patients undergoing oncologic [^18^F]FDG-PET/CT reported that PS and SUV reconstructions were subjectively of comparable quality, though PS images were again derived from earlier post-injection time points than SUV images [8]. These same two studies also reported that Patlak-derived reconstructions (including PS images) may occasionally identify malignant lesions not seen on SUV images or allow lesions that appear suspicious on SUV images to be dismissed as benign [8,9]. Although our study did not entail the use of a reference standard to compare the diagnostic accuracy of PS versus SUV images, we did find that late PS images allow for the identification of a slightly higher number of tracer-avid lesions than early SUV images but similar to slightly fewer tracer-avid lesions than late SUV images.

Regarding quantitative assessments of lesion conspicuity, which focus specifically on liver lesions, we found that PS images provide a higher TBR than SUV images at early post-injection time points, with opposite findings at late post-injection time points. Prior studies reported higher TBRs for PS images than for SUV images, though this finding was confounded by the different uptake times for SUV and PS images [8,9]. We observed that the benefits of PS images in terms of hepatic lesion conspicuity are highly dependent on the uptake times; PS images are unlikely to offer much (if any) added value for hepatic lesion detection relative to SUV images, provided that the PET data are acquired after a sufficiently long delay (75 min in our study). For SUV images, the higher late hepatic TBRs reflect continuous accumulation of tracer by malignant lesions during the post-injection period; significant early-to-late decreases in background hepatic tracer activity also contributed to this finding for the [^18^F]FDG subgroup, likely reflecting hepatic [^18^F]FDG efflux related to physiological dephosphorylation [15,16].

Our study has several limitations. First, there may be biases related to our single-center design and utilization of a single PET scanner. Our results should be confirmed at other institutions and on other PET scanner models. Due to the heterogeneity of the patient cohort and the relatively small sample size, it was not possible to perform subgroup analysis based on particular tumor types or clinical indications. The PET and PS reconstructions utilized parameters specifically recommended by the scanner’s manufacturer. We did not independently optimize these settings for image quality. Therefore, the observed qualitative inferiority of the PS-early images might be overcome by future modifications to the reconstruction parameters. Our study also did not utilize a reference standard to adjudicate the diagnostic accuracy of PS versus SUV images. Larger studies focusing on particular cancer types and imaging indications will be needed to achieve sufficient power to address questions of clinical impact. Due to our relatively small sample sizes, cases/lesions were pooled across tracer types for some analyses, though tracer-specific analyses were also performed to evaluate for any effects related to differences in tracer behavior. Finally, our study excluded subjects who reported potential difficulties tolerating a 90-min imaging period. Although data acquisition for PS reconstruction can be accomplished much faster than in our study (i.e., only three WB passes), the recruitment strategy may have enriched our cohort for patients capable of remaining relatively motionless during imaging. Consequently, PS images, the reconstruction of which is based on the expectation that patients remain nearly motionless throughout the dynamic PET imaging period, may be of lower quality in an unselected oncologic population.

## 5. Conclusions

In conclusion, late PS images were similar to early/late SUV images in terms of image quality and lesion detection, whereas early PS images were of inferior image quality. Among tracer-avid liver lesions, early PS images had higher tumor-to-background ratios compared with early SUV images, whereas late PS images had lower tumor-to-background ratios compared with late SUV images.

## Figures and Tables

**Figure 1 diagnostics-14-00883-f001:**
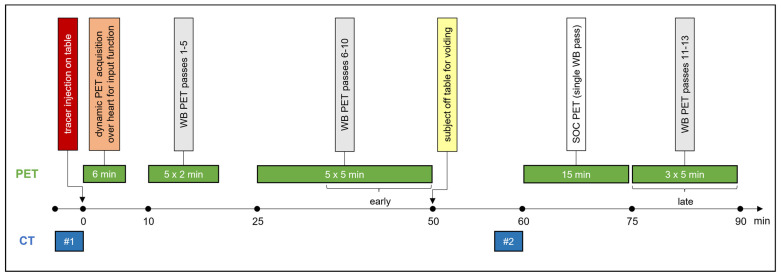
PET/CT image acquisition protocol. Brackets indicate portions of PET data utilized for early and late PS and SUV reconstructions. Abbreviations: CT = computed tomography; PET = positron emission tomography; SOC = standard of care; WB = whole-body.

**Figure 2 diagnostics-14-00883-f002:**
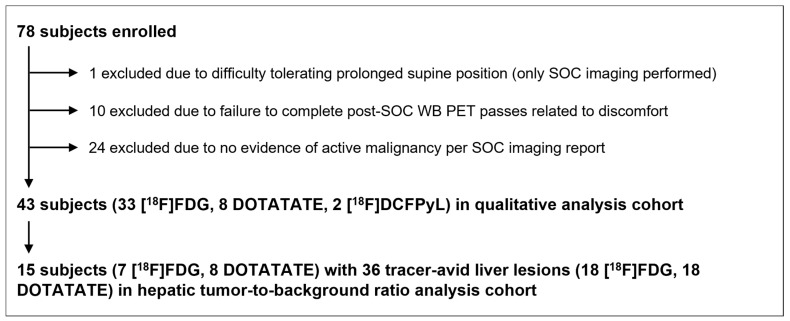
Study flowchart. Abbreviations: PET = positron emission tomography; SOC = standard of care; WB = whole-body.

**Figure 3 diagnostics-14-00883-f003:**
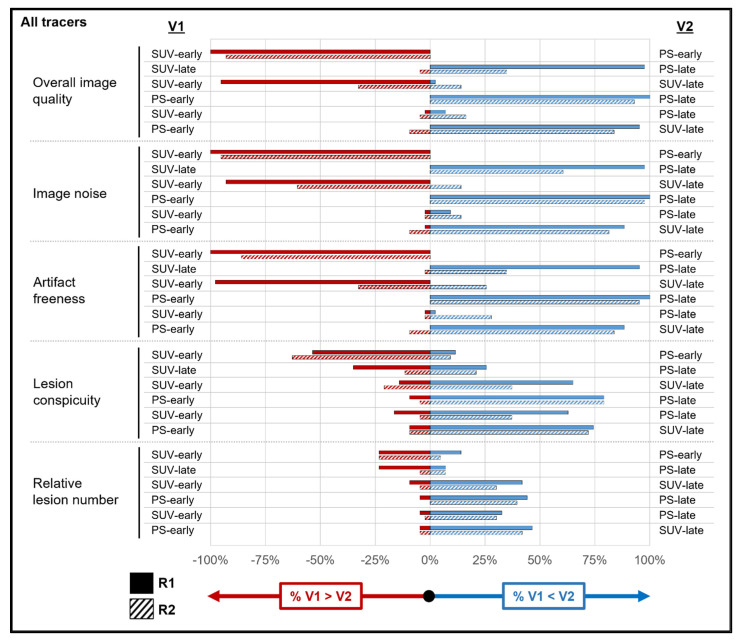
Reader reconstruction preferences in qualitative analysis across all tracers. For each image feature assessed by the readers in the qualitative analysis (including all participants, regardless of tracer), the horizontal bars show the percentage of cases in which a given reader preferred one reconstruction versus another. Red bars reflect the percentage of cases in which the reconstruction in the variable 1 (V1) column was preferred (by any magnitude) over the reconstruction in the variable 2 (V2) column. Blue bars reflect the percentage of cases in which the reconstruction in the V1 column was preferred (by any magnitude) over the reconstruction in the V2 column. Cases in which a reader expressed no preference for a given pair of reconstructions are not shown. Thus, short red and/or blue bars reflect the greater similarity between the V1 and V2 column reconstructions, whereas long red and/or blue bars reflect large preferences for the V1 (red bars) or V2 (blue bars) column over the other. See Appendix A for source data. Abbreviations: PS = Patlak slope; R1 = reader 1; R2 = reader 2; SUV = standardized uptake value.

**Figure 4 diagnostics-14-00883-f004:**
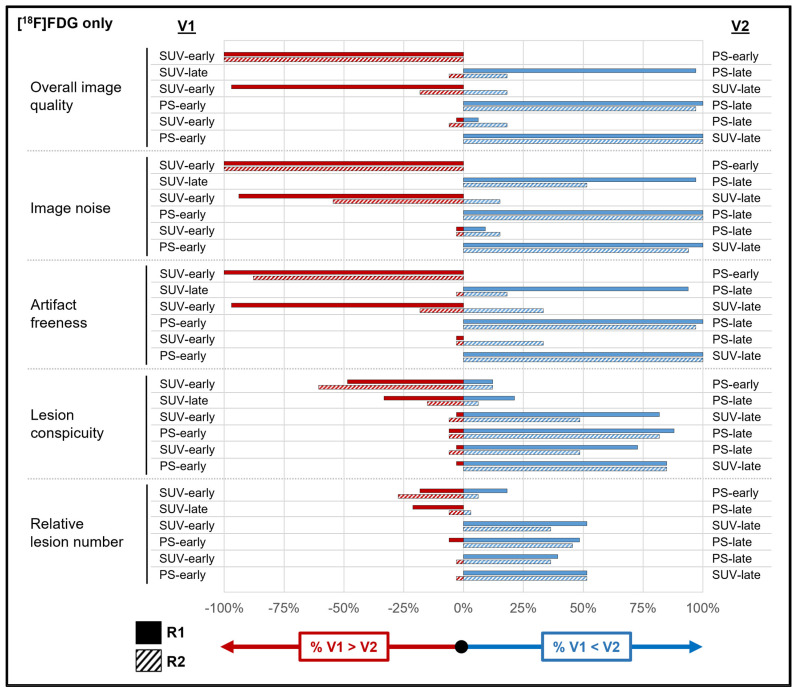
Reader reconstruction preferences in qualitative analysis for [^18^F]FDG only. See Figure 3 legend for interpretive guidance. See Appendix A for source data. Abbreviations: PS = Patlak slope; R1 = reader 1; R2 = reader 2; SUV = standardized uptake value; V1 = variable 1; V2 = variable 2.

**Figure 5 diagnostics-14-00883-f005:**
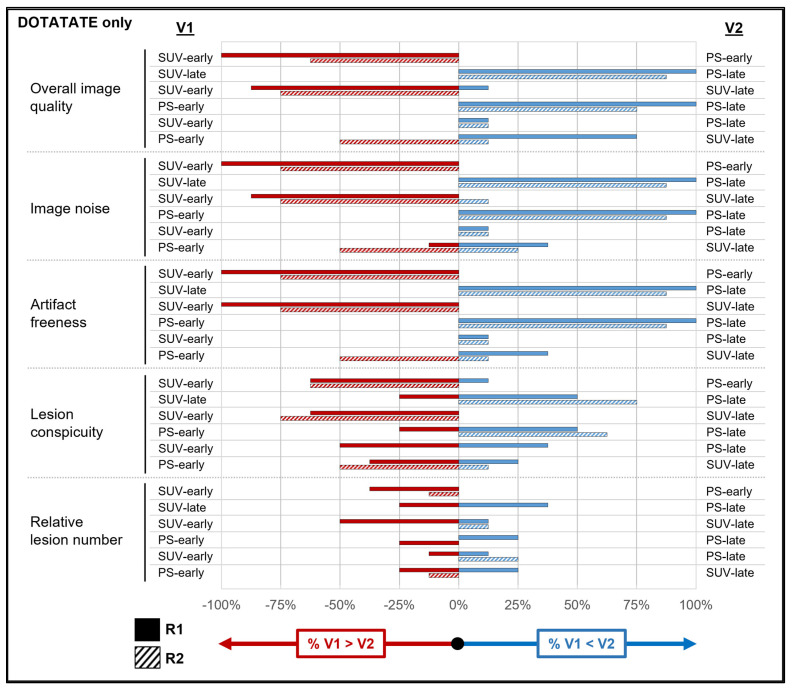
Reader reconstruction preferences in qualitative analysis for DOTATATE only. See Figure 3 legend for interpretive guidance. See Appendix A for source data. Abbreviations: PS = Patlak slope; R1 = reader 1; R2 = reader 2; SUV = standardized uptake value; V1 = variable 1; V2 = variable 2.

**Figure 6 diagnostics-14-00883-f006:**
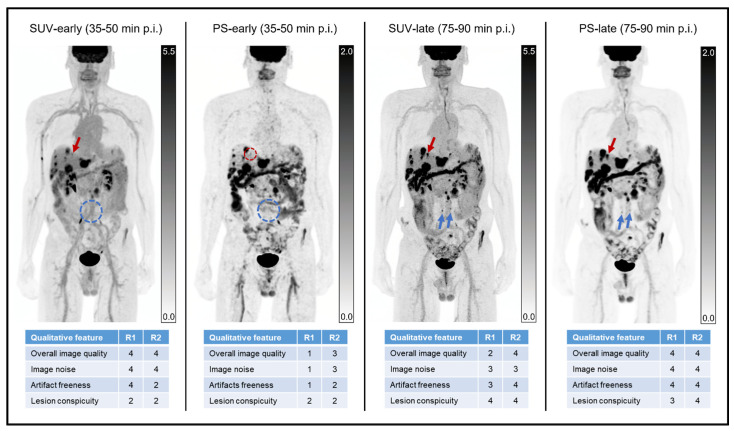
Example of qualitative image quality across reconstructions. A 67-year-old man on systemic therapy for lymphoma underwent a restaging [^18^F]FDG-PET/CT. Maximum intensity projection coronal standardized uptake value (SUV) and Patlak slope (PS) images are shown at early (35–50 min p.i.) and late (75–90 min p.i.) time points, along with corresponding Likert assessments of qualitative imaging features for both readers. As reflected in these images (as well as in Appendix A), the PS-early reconstruction generally had higher noise, more artifacts, lower lesion conspicuity, and lower overall image quality than the other reconstructions for both readers. As reflected in the reader scores for this case, the SUV-early and PS-late reconstructions were generally scored as having better overall image quality than the SUV-late and PS-early images. Both readers noted the fewest number of lesions for the PS-early reconstruction, with additional lesions seen in the liver and/or lymph nodes for the other reconstructions. Note that at least one [^18^F]FDG-avid liver lesion seen on three of the reconstructions (red arrows) is not readily appreciated on the PS-early images (dashed red circle). Similarly, several [^18^F]FDG-avid retroperitoneal lymph nodes seen on the SUV-late and PS-late reconstructions (blue arrows) were not apparent on the SUV-early or PS-early reconstructions (dashed blue circles). In general (see Appendix A), the PS-early reconstruction showed fewer lesions than the other three reconstructions.

**Figure 7 diagnostics-14-00883-f007:**
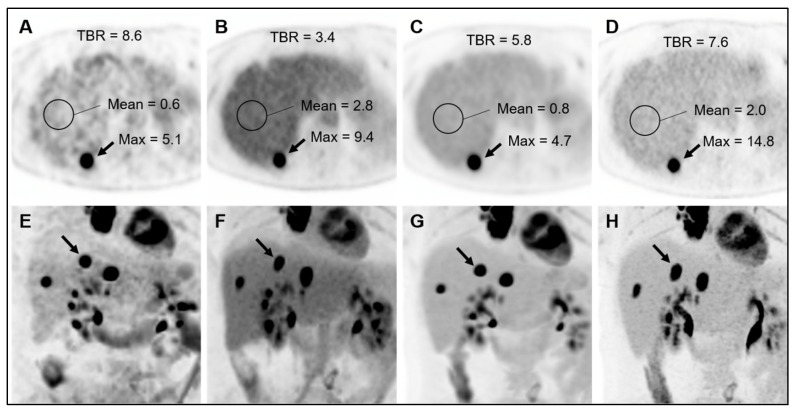
Example of differences in tumor-to-background ratios across reconstructions. A 58-year-old woman underwent initial staging [^18^F]FDG-PET/CT. Axial PET (**A**–**D**) and coronal maximum intensity projection PET (**E**–**H**) images are shown for PS-early (**A**,**E**), SUV-early (**B**,**F**), PS-late (**C**,**G**), and SUV-late (**D**,**H**) reconstructions. Maximum values of an [^18^F]FDG-avid metastasis (arrows) in hepatic segment 7 are provided. The mean values of the background liver parenchyma are also shown. As was generally observed (averaged across all liver lesions), the tumor-to-background ratio (TBR) was higher on PS images than on SUV images at the early time point. In contrast, the TBR was higher on SUV images than on PS images at the late time point.

**Table 1 diagnostics-14-00883-t001:** Participant and scan characteristics.

Characteristic	Value
Age (years)—mean ± standard deviation	63.3 ± 8.3
Sex—n (%)	
Male	26 (60.5)
Female	17 (39.5)
Non-binary	0 (0.0)
Cancer type—n (%)	
Head/neck SCC	1 (2.3)
Thyroid	2 (4.7)
Esophageal	1 (2.3)
Lung	7 (16.2)
Breast	2 (4.7)
Melanoma	3 (7.0)
Gastric	1 (2.3)
Pancreas	1 (2.3)
Colorectal	4 (9.3)
Lymphoma	3 (7.0)
Neuroendocrine	9 (21.0)
Cutaneous SCC	1 (2.3)
Ovarian	1 (2.3)
Cervical	2 (4.7)
Fallopian	1 (2.3)
Prostate	2 (4.7)
Bladder	1 (2.3)
Leiomyosarcoma	1 (2.3)
Indication for PET—n (%)	
Diagnosis of suspected malignancy	2 (4.7)
Initial staging of confirmed malignancy	2 (4.7)
Restaging during/after treatment	29 (67.4)
Detection of a suspected recurrence	10 (23.2)
Surveillance	0 (0.0)
Tracer—n (%)	
[^18^F]FDG	33 (76.7)
[^68^Ga]Ga-DOTATATE	6 (14.0)
[^64^Cu]Cu-DOTATATE	2 (4.7)
[^18^F]DCFPyL	2 (4.7)
Number of lesions analyzed per participant—n (%)	
1 lesion	8 (18.6)
2 lesions	13 (30.2)
3 lesions	9 (21.0)
4 lesions	7 (16.2)
5 lesions	6 (14.0)

Abbreviations: PET = positron emission tomography; SCC = squamous cell carcinoma.

**Table 2 diagnostics-14-00883-t002:** Hepatic tumor-to-background ratios and background liver temporal stability of PS and SUV metrics.

Variable 1 (V1)	V1 Median (Q1, Q3)	Variable 2 (V2)	V2 Median (Q1, Q3)	*p* Value (V1 vs. V2)
PS-early-max TBR		PS-late-max TBR		
All tracers	3.87 (2.79, 7.35)	All tracers	3.57 (2.31, 5.24)	<0.001
[^18^F]FDG only	3.89 (2.83, 6.78)	[^18^F]FDG only	4.05 (2.33, 5.17)	0.005
DOTATATE only	3.87 (2.92, 8.95)	DOTATATE only	3.52 (2.42, 7.93)	0.25
PS-early-peak TBR		PS-late-peak TBR		
All tracers	2.90 (2.02, 5.69)	All tracers	2.80 (1.73, 4.43)	0.03
[^18^F]FDG only	2.76 (1.89, 5.42)	[^18^F]FDG only	2.86 (1.71, 4.29)	0.06
DOTATATE only	3.09 (2.11, 6.87)	DOTATATE only	2.65 (2.00, 6.72)	0.16
SUV-early-max TBR		SUV-late-max TBR		
All tracers	3.09 (2.10, 4.16)	All tracers	5.29 (3.41, 7.45)	<0.001
[^18^F]FDG only	2.45 (1.74, 3.34)	[^18^F]FDG only	4.79 (2.96, 6.79)	<0.001
DOTATATE only	3.87 (3.05, 10.87)	DOTATATE only	5.37 (3.55, 12.71)	0.002
SUV-early-peak TBR		SUV-late-peak TBR		
All tracers	2.28 (1.61, 3.38)	All tracers	3.10 (2.01, 4.48)	<0.001
[^18^F]FDG only	1.83 (1.43, 2.4)	[^18^F]FDG only	3.26 (1.81, 4.4)	<0.001
DOTATATE only	2.94 (2.20, 7.4)	DOTATATE only	2.95 (2.18, 7.50)	0.20
PS-early-max TBR		SUV-early-max TBR		
All tracers	3.87 (2.79, 7.35)	All tracers	3.09 (2.1, 4.16)	0.006
[^18^F]FDG only	3.89 (2.83, 6.78)	[^18^F]FDG only	2.45 (1.74, 3.34)	<0.001
DOTATATE only	3.87 (2.92, 8.95)	DOTATATE only	3.87 (3.05, 10.87)	0.25
PS-early-peak TBR		SUV-early-peak TBR		
All tracers	2.90 (2.02, 5.69)	All tracers	2.28 (1.61, 3.38)	0.003
[^18^F]FDG only	2.76 (1.89, 5.42)	[^18^F]FDG only	1.83 (1.43, 2.4)	0.003
DOTATATE only	3.09 (2.11, 6.87)	DOTATATE only	2.94 (2.20, 7.40)	0.56
PS-late-max TBR		SUV-late-max TBR		
All tracers	3.57 (2.31, 5.24)	All tracers	5.29 (3.41, 7.45)	<0.001
[^18^F]FDG only	4.05 (2.33, 5.17)	[^18^F]FDG only	4.79 (2.96, 6.79)	<0.001
DOTATATE only	3.52 (2.42, 7.93)	DOTATATE only	5.37 (3.55, 12.71)	<0.001
PS-late-peak TBR		SUV-late-peak TBR		
All tracers	2.80 (1.73, 4.43)	All tracers	3.10 (2.01, 4.48)	<0.001
[^18^F]FDG only	2.86 (1.71, 4.29)	[^18^F]FDG only	3.26 (1.81, 4.4)	0.01
DOTATATE only	2.65 (2.00, 6.72)	DOTATATE only	2.95 (2.18, 7.50)	<0.001

Abbreviations: Q1 = first quartile; Q3 = third quartile; TBR = tumor-to-background ratio.

## Data Availability

The data presented in this study are available upon request from the corresponding author. The data are not publicly available due to restrictions aimed at protecting patient confidentiality.

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
