# Peer review of "Patlak Slope versus Standardized Uptake Value Image Quality in an Oncologic PET/CT Population: A Prospective Cross-Sectional Study"

_diagnostics, 2024, doi:10.3390/diagnostics14090883_

Round 1

Reviewer 1 Report

Comments and Suggestions for Authors

Thank you very much for the opportunity to review the manuscript entitled “Patlak Slope Versus Standardized Uptake Value Image Quality in an Oncologic PET/CT Population: A Prospective Cross-sectional Study”. The manuscript is interesting, however requires a majority of changes before considering for the publication.

1.       Please provide changes of the nomenclature in whole manuscript according to "Consensus nomenclature rules for radiopharmaceutical chemistry — setting the record straight"

2.       Abstract section: please explain all used abbreviations in this section.

3.       Introduction section: please explain abbreviations for used radiopharmaceuticals.

4.       Material and Methods section: please explain all used abbreviations in this section.

5.       Material and Methods, line 72: abbreviation SOC should be explained earlier in the line 65, where is first used in the main text.

6.       Figure 1 – please explain all used abbreviation – for example PET and CT are not explained.

7.       Material and Methods, page 3, line 119-120: the same sentence is provided already in the line 106-108.

8.       Table 1: 9 patients had a neuroendocrine tumors, while authors wrote that in 8 patients DOTATATE which is designed for neuroendocrine tumors was used. Thus, does it mean that one patient undergo FDG or DCFPyL study?

9.       Table 1: please explain all used abbreviations in this table.

10.   Figure 2: for TBR only 15 patients were included with two different tracers which also show different biodistribution, especially in liver – DOTATETE shows high uptake in the liver, while FDG lower, thus combining these two different tracers and comparing values should not be performed. Moreover, this is a relatively small group of patients, and this might have an influence on the obtained results.

Author Response

Manuscript ID: diagnostics-2931762

Thank you for the opportunity to submit a revised version of the manuscript referenced above. Please see responses to individual reviewer comments below:

REVIEWER 1

Thank you very much for the opportunity to review the manuscript entitled “Patlak Slope Versus Standardized Uptake Value Image Quality in an Oncologic PET/CT Population: A Prospective Cross-sectional Study”. The manuscript is interesting, however requires a majority of changes before considering for the publication.

  1. Please provide changes of the nomenclature in whole manuscript according to "Consensus nomenclature rules for radiopharmaceutical chemistry — setting the record straight"

RESPONSE: We have changed all radiopharmaceutical names to be consistent with this consensus nomenclature.

  1. Abstract section: please explain all used abbreviations in this section.

RESPONSE: All abbreviations are now defined at first use.

  1. Introduction section: please explain abbreviations for used radiopharmaceuticals.

RESPONSE: All abbreviations are now defined at first use.

  1. Material and Methods section: please explain all used abbreviations in this section.

RESPONSE: All abbreviations are now defined at first use.

  1. Material and Methods, line 72: abbreviation SOC should be explained earlier in the line 65, where is first used in the main text.

RESPONSE: All abbreviations are now defined at first use.

  1. Figure 1 – please explain all used abbreviation – for example PET and CT are not explained.

RESPONSE: We now define all abbreviations in the Figure and Table legends.

  1. Material and Methods, page 3, line 119-120: the same sentence is provided already in the line 106-108.

RESPONSE: We have deleted the redundant material in the paragraph referenced by the reviewer.

  1. Table 1: 9 patients had a neuroendocrine tumors, while authors wrote that in 8 patients DOTATATE which is designed for neuroendocrine tumors was used. Thus, does it mean that one patient undergo FDG or DCFPyL study?

RESPONSE: Yes, one patient with a neuroendocrine tumor was imaged with FDG due to high-grade disease.

  1. Table 1: please explain all used abbreviations in this table.

RESPONSE: We now define all abbreviations in the Figure and Table legends.

  1. Figure 2: for TBR only 15 patients were included with two different tracers which also show different biodistribution, especially in liver – DOTATETE shows high uptake in the liver, while FDG lower, thus combining these two different tracers and comparing values should not be performed. Moreover, this is a relatively small group of patients, and this might have an influence on the obtained results.

RESPONSE: We added the following text to the Discussion section to highlight the limitations related to pooling across tracer types: “Due to our relatively small sample sizes, cases/lesions were pooled across tracer types for some analyses, though tracer-specific analyses were also performed to evaluate for any effects related to differences in tracer behavior.”

Reviewer 2 Report

Comments and Suggestions for Authors

Manuscript title "Patlak Slope Versus Standardized Uptake Value Image Quality in an Oncologic PET/CT Population: A Prospective Cross-sectional Study"

1. The authors present a prospective study evaluating two paired image sets at two time points in oncological PET/CT.

2. The study is original and relevant to the field of oncological imaging. The authors may consider providing the relevance of false-negative results for the PERCIST criteria.

3. The main study strengths include good rationale, extensive methodology, excellent figures. All this allows for a high degree of confidence in result reproducibility. The main study limitations are small sample size without sample size estimation and "pooling" of various tracers.

4. The conclusions are consistent with the evidence provided.

5. The references are appropriate.

6. Figures and tables are informative.

7. Ethics and data availability statements are adequate.

Author Response

Manuscript ID: diagnostics-2931762

Thank you for the opportunity to submit a revised version of the manuscript referenced above. Please see responses to individual reviewer comments below:

REVIEWER 2

  1. The authors present a prospective study evaluating two paired image sets at two time points in oncological PET/CT.
  2. The study is original and relevant to the field of oncological imaging. The authors may consider providing the relevance of false-negative results for the PERCIST criteria.

RESPONSE: We do not understand the relevance to our manuscript of this comment about false-negative results for the PERCIST criteria. If the review could elaborate on this comment and provide a relevant line/page number, we would be happy to reconsider a change.

  1. The main study strengths include good rationale, extensive methodology, excellent figures. All this allows for a high degree of confidence in result reproducibility. The main study limitations are small sample size without sample size estimation and "pooling" of various tracers.

RESPONSE: We added the following text to the Discussion section to highlight the limitations related to pooling across tracer types: “Due to our relatively small sample sizes, cases/lesions were pooled across tracer types for some analyses, though tracer-specific analyses were also performed to evaluate for any effects related to differences in tracer behavior.”

  1. The conclusions are consistent with the evidence provided.
  2. The references are appropriate.
  3. Figures and tables are informative.
  4. Ethics and data availability statements are adequate.

Round 2

Reviewer 1 Report

Comments and Suggestions for Authors

Thank you very much for providing the requested changes. No further changes are required.